# Spectral Probe for Electron Transfer and Addition Reactions of Azide Radicals with Substituted Quinoxalin-2-Ones in Aqueous Solutions

**DOI:** 10.3390/ijms22020633

**Published:** 2021-01-10

**Authors:** Konrad Skotnicki, Slawomir Ostrowski, Jan Cz. Dobrowolski, Julio R. De la Fuente, Alvaro Cañete, Krzysztof Bobrowski

**Affiliations:** 1Centre of Radiation Research and Technology, Institute of Nuclear Chemistry and Technology, 03-195 Warsaw, Poland; 2Centre of Radiochemistry and Nuclear Chemistry, Institute of Nuclear Chemistry and Technology, 03-195 Warsaw, Poland; s.ostrowski@ichtj.waw.pl (S.O.); j.dobrowolski@ichtj.waw.pl (J.C.D.); 3Departamento de Química Organica y Fisicoquímica, Facultad de Ciencias Químicas y Farmaceuticas, Universidad de Chile, Casilla 223, Santiago 8380492, Chile; jrfuente@ciq.uchile.cl; 4Instituto de Ciencias Químicas Aplicadas, Universidad Autónoma de Chile, Santiago 8380492, Chile; alvcanete67@gmail.com

**Keywords:** azide radical, one-electron radical oxidants, electron transfer, addition, quinoxalin-2-ones, pulse radiolysis, DFT and TD-DFT calculations

## Abstract

The azide radical (N_3_^●^) is one of the most important one-electron oxidants used extensively in radiation chemistry studies involving molecules of biological significance. Generally, it was assumed that N_3_^●^ reacts in aqueous solutions only by electron transfer. However, there were several reports indicating the possibility of N_3_^●^ addition in aqueous solutions to organic compounds containing double bonds. The main purpose of this study was to find an experimental approach that allows a clear assignment of the nature of obtained products either to its one-electron oxidation or its addition products. Radiolysis of water provides a convenient source of one-electron oxidizing radicals characterized by a very broad range of reduction potentials. Two inorganic radicals (SO_4_^●−^, CO_3_^●−^) and Tl^2+^ ions with the reduction potentials higher, and one radical (SCN)_2_^●−^ with the reduction potential slightly lower than the reduction potential of N_3_^●^ were selected as dominant electron-acceptors. Transient absorption spectra formed in their reactions with a series of quinoxalin-2-one derivatives were confronted with absorption spectra formed from reactions of N_3_^●^ with the same series of compounds. Cases, in which the absorption spectra formed in reactions involving N_3_^●^ differ from the absorption spectra formed in the reactions involving other one-electron oxidants, strongly indicate that N_3_^●^ is involved in the other reaction channel such as addition to double bonds. Moreover, it was shown that high-rate constants of reactions of N_3_^●^ with quinoxalin-2-ones do not ultimately prove that they are electron transfer reactions. The optimized structures of the radical cations (7-R-3-MeQ)^●+^, radicals (7-R-3-MeQ)^●^ and N_3_^●^ adducts at the C2 carbon atom in pyrazine moiety and their absorption spectra are reasonably well reproduced by density functional theory quantum mechanics calculations employing the ωB97XD functional combined with the Dunning’s aug-cc-pVTZ correlation-consistent polarized basis sets augmented with diffuse functions.

## 1. Introduction

The azide radical (N_3_^●^) is one of the most important one-electron oxidants used extensively in radiation chemistry studies involving inorganic [1,2,3], and aromatic compounds [4,5,6,7,8,9], and also molecules of biological significance [10,11,12,13,14,15]. In general, it is assumed that N_3_^●^ reacts in aqueous solution only by electron transfer [4,16,17]. Therefore, the value of the standard reduction potential of the N_3_^●^/N_3_^−^ redox couple (E^0^ = 1.33 ± 0.01 V vs. NHE) [18], which applies to reactions in aqueous solutions, is essential for understanding the mechanism and the kinetics of oxidation of organic compounds by N_3_^●^. Its oxidation reactions are particularly rapid, even more rapid than the reactions involving some stronger oxidants such as Br_2_^●−^ and CO_3_^●−^ [19]. This is probably due to the high self-exchange rate constant of ≈4 × 10^4^ M^−1^ s^−1^ for the N_3_^●^/N_3_^−^ couple inferred from the cross-relationship of Marcus theory [16].

Oxidation of aniline and its N-methyl derivatives to the respective radical cations (C_6_H_5_NH_2_^●+^, C_6_H_5_NH(CH_3_)^●+^, C_6_H_5_N(CH_3_)_2_^●+^) [5], phenol and its substituted methyl and methoxy derivatives to the respective phenoxyl radicals (PhO^●^, CH_3_PhO^●^ and CH_3_OPhO^●^) [4], indole and its substituted hydroxy (5,6-dihydroxyindole (DHI)) and methoxy (5,6-dimethoxyindole (DMI)) derivatives to the respective indolyl radical cations (In^●+^, DHI^●+^, DMI^●+^) or indolyl radicals [10,11], and thiols (RSH) to the respective thiyl radicals (RS^●^) [12], are the very well documented examples of oxidation reactions involving N_3_^●^.

The phenol, indole, and thiol structures are ubiquitous in biology. Of prime importance are tyrosine (TyrOH), tryptophan (TrpNH) and cysteine (CysSH) amino acids which were in the past the subjects to numerous N_3_^●^-induced one-electron oxidation studies in peptides [20,21,22,23,24], and proteins [25,26,27,28,29,30]. The N_3_^●^ react with TyrOH, TrpNH, and CysSH (Equations (1)–(3)) with the rate constants 1.0 × 10^8^ M^−1^s^−1^ [25], 4.1 × 10^9^ M^−1^s^−1^ [25,31], and 1.4 × 10^7^ M^−1^s^−1^ [25], respectively:N_3_^●^ + TyrOH → N_3_^−^ + TyrOH^●+^ → N_3_^−^ + TyrO^●^ + H^+^(1)
N_3_^●^ + TrpNH → N_3_^−^ + TrpNH^●+^ → N_3_^−^ + TrpN^●^ + H^+^(2)
N_3_^●^ + CySH → N_3_^−^ + CysSH^●+^ → N_3_^−^ + CysS^●^ + H^+^(3)

These reactions lead to tyrosyl (TyrO^●^), tryptophyl (TrpN^●^) and cysteine thiyl (CysS^●^) radicals which can be easily identified based on their absorption spectra. The rates of one-electron oxidation of deprotonated tyrosine (TyrO^−^) and cysteine (CysS^−^) were found to be higher by one or even two orders of magnitude than those of the protonated species: *k*_N3● + TyrO−_ = 3.6 × 10^9^ M^−1^ s^−1^ and *k*_N3● + CysS−_ = 2.7 × 10^9^ M^−1^ s^−1^ [25]. This fact was rationalized on the basis of the higher electron-donating ability of the TyrO^−^ and CysS^−^ anions.

Other interesting examples of using N_3_^●^ as one-electron oxidant were the studies involving sulfur-substituted nucleobases (thiobases). It was shown that the reactions of 4-thiouracil (4TU(=S)) [32], and 2-thiouracil (2TU(=S)) [15] with N_3_^●^ led to the respective dimeric 2c-3e S-S-bonded radicals in their cationic and neutral forms, respectively (Equations (4) and (5)).
N_3_^●^ + 4TU(=S) → N_3_^−^ + 4TU^+^(S^●^)4TU^+^(S^●^) + 4TU(=S) ⇆ (4TUS∴S4TU)^+^(4)
N_3_^●^ + 2TU(=S) → N_3_^−^ + 2TU^+^(S^●^)2TU^+^(S^●^) → 2TUS^●^ + H^+^2TU^+^(S^●^) + 2TU(=S) ⇆ (2TUS∴S2TU)(5)

These observations are in line with the lower reduction potential of 2-TU (≈ +0.7 V vs. NHE) in comparison to N_3_^●^ and therefore in such designed systems, oxidation of 2-TU leads directly to 2TU^+^(S^●^) radical cations.

Interestingly, there are several reports indicating the possibility of N_3_^●^ addition in aqueous solutions to organic compounds containing double bonds. Spin-trapping experiments performed for the N_3_^●^ detection in aqueous solutions of an azide/catalase/H_2_O_2_ and an azide/peroxidase/H_2_O_2_ using phenyl-tert-butyl nitrone (PBN) and 5,5-dimetyl-1-pyrroline-N-oxide (DMPO) confirmed the presence PBN/DMPO-N_3_ radical adducts which were detected by ESR techniques. They are formed via reactions represented by Equations (6) and (7), respectively. By using ^14^N- and ^15^N-labelled NaN_3_ it was possible to confirm unequivocally that N_3_^●^ added to the C=N bond in PBN [33].

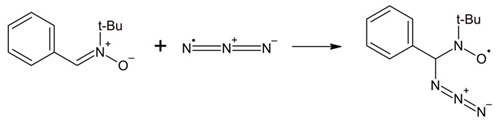
(6)

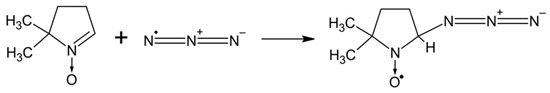
(7)

PBN-N_3_ radical adducts were detected in similar spin-trapping experiments performed on photolyzed aqueous solutions containing azide/H_2_O_2_ [34], and azido cobalt(III) complexes [35]. Similarly, DMPO-N_3_ radical adducts were also detected in aqueous solutions of azide/cytochrome c oxidase and azide/cytochrome c oxidase/H_2_O_2_ systems [36], azide/H_2_O_2_-activated endogeneous cytochrome c peroxidase [37], and in a photolyzed azide/H_2_O_2_ system [34]. These experiments clearly showed the possibility for another reaction channel of N_3_^●^ in aqueous solutions such as addition to double bonds (Equations (6) and (7)). However, they do not provide us with information related to the reactivity of N_3_^●^ by processes other than electron transfer. Such knowledge is important for evaluating the usefulness of N_3_^●^ as a secondary oxidant in biological studies.

It should be noted that the standard reduction potential of the N_3_^●^/N_3_^−^ redox couple decreases in going from polar to less polar solvents [38]. Therefore N_3_^●^ is not as strong oxidizing agent in such solvents as it is in water. This leaves open the possibility for other reaction channels such as addition to double bonds or hydrogen abstraction. For instance, the absolute rate constants for the addition reactions of N_3_^●^ with a series of ring-substituted styrenes (*p*-CF_3_, *m*-CF_3_, *p*-Cl, H, *m*-CH_3_, *p*-CH_3_, *p*-CH_3_O) in acetonitrile were found to vary between 1 × 10^6^ M^−1^ s^−1^ and 5 × 10^7^ M^−1^ s^−1^. A correlation of log(*k*_add_) with Hammett σ^+^ constants yields ρ^+^ = −1.2 which indicates the electrophilic nature of N_3_^●^ [39]. On the other hand, the rate constants for the reaction of N_3_^●^ with α- and β-substituted styrenes and simple alkyl- and alkoxy-substituted olefins which vary between 1 × 10^6^ M^−1^ s^−1^ and 1 × 10^9^ M^−1^ s^−1^ were nicely correlated with the corresponding ionization potentials (IP_e_). The negative slope of the linear plot of log(*k*_add_) versus IP_e_ indicates that these reactions are occurring with considerable charge transfer interactions in the transition state and are dominated by polar effects [39].

With these premises, in the current studies we report our investigations on the reaction of N_3_^●^ with the quinoxalin-2-one derivatives which are multifunctional molecules consisting of an aromatic ring connected in neighboring positions with a heterocyclic ring, in aqueous solutions. We selected six 3-methyl-1H-quinoxalin-2-one derivatives, five of them substituted in position 7 with various electron donating (-OCH_3_, -NH_2_) and electron withdrawing (-F, -CN, -CF_3_) groups (Figure 1).

Quinoxalin-2-one derivatives are also ubiquitous in biology and have recently received much attention connected with their biological properties and pharmaceutical applications [40,41]. A key factor that is decisive in their biological activity is substitution at the carbon-3 in the pyrazine ring and at carbons 6 and/or 7 in the benzene ring. Nearly all biologically active derivatives are substituted in those specific positions. The application of quinoxaline derivatives is strongly related to possible one-electron redox processes involving the quinoxaline moiety. There are several comprehensive reviews covering quinoxaline chemistry and applications [40,42,43,44,45,46,47]. Nonetheless, studies devoted to quinoxaline derived radicals are rather scarce and include studies performed by pulse radiolysis [48,49,50,51,52], photochemistry [53,54,55,56,57], electrochemistry [58,59,60], and Fenton systems [61].

In principle, it is well known that reduction potentials are linearly dependent on Brown’s σ_p_^+^ values as it was shown for benzene radical cations [62], and 4-substituted aniline radical cations [63]. In other words, electron donating substituents lower the one-electron reduction potential of a given redox couple. Taking into account Brown’s σ_p_^+^ values [64], one can expect the highest reduction potential for the 7-CN-3MeQ derivative and the lowest reduction potential for the 7-NH_2_-3-MeQ derivative. This conclusion is also in line with reduction potential predictions, based on accurate quantum chemical methods (DFT), of quinoxaline and a number of its derivatives with electron-donating and electron-withdrawing substituent groups [65].

Structure Activity Relationship (SAR) studies revealed that quinoxalin-2-ones derivatives are bound in very specific positions in proteins [66,67]. Therefore, they or radicals derived from them can interact with either amino acid residues or radicals derived from them. For example, certain amino acid residues—TyrOH, TrpNH, and CysSH—are particularly vulnerable to oxidation. Therefore, the radical cations derived from quinoxalin-2-one derivatives can potentially oxidize them to TyrO^●^, TrpN^●^ and CysS^●^ radicals, respectively. On the other hand, these radicals are reasonably good electron acceptors and can act as oxidants of quinoxalin-2-one derivatives intercalated in a protein matrix.

These examples strongly indicate a need to obtain a comprehensive and systematized, and at the same time reliable knowledge about spectral and kinetic properties of radicals and radical ions derived from quinoxalin-2-ones. Since N_3_^●^ is both commonly used as an one-electron oxidant of biologically important compounds, and yet has the confirmed possibility of adding to double bonds, there is a strong need to unambiguously assign observed transient absorption spectra to either its one-electron oxidation products or its addition products. Therefore, the main aim of this current work was to find an experimental approach that would allow us to clearly define the nature of the obtained products.

Radiolysis of water provides a very convenient source of one-electron oxidizing radicals characterized by a very broad range of reduction potentials which is very useful for studying oxidation reactions of molecules of biological significance [68,69]. In the current study, we selected two inorganic radicals (SO_4_^●−^, CO_3_^●−^) and Tl^2+^ ions, with reduction potentials higher than the reduction potential of N_3_^●^, and one radical (SCN)_2_^●−^ with a nearly equal, however lower, reduction potential as that of N_3_^●^ (vide Appendix A). These radicals and Tl^2+^ ions act predominantly as electron acceptors, however, the radicals can also react by hydrogen abstraction or addition reactions [70,71,72,73,74,75,76,77]. These latter reactions are generally very slow and might not be observed by pulse radiolysis [19].

The basic idea of this experimental study is that we have probed absorption spectra formed in reactions of one-electron oxidants with a series of quinoxalin-2-one derivatives and have confronted these spectra with absorption spectra formed from reactions of azide radicals with the same series of compounds. The mere fact that reactions of N_3_^●^ with quinoxalin-2-ones take place with high rate constants does not ultimately prove that they are electron transfer reactions [50]. Cases, in which the absorption spectra formed in reactions involving N_3_^●^ differ from the absorption spectra formed in the reactions involving other one-electron oxidants, may indicate that N_3_^●^ does not act as electron acceptor.

## 2. Results

### 2.1. Reactions of SO_4_^●−^ and N_3_^●^ with 3-Methyl-1H-Quinoxalin-2-One (3-MeQ) and Its 7-substituted Derivatives (7-R-3MeQ)–Absorption Spectra

#### 2.1.1. Absorption Spectra Recorded after Reaction of N_3_^●^ and SO_4_^●−^ with 3-MeQ

At the beginning, we selected for comparative studies the sulfate radical anion (SO_4_^●−^) with the highest reduction potential (E^0^ = +2.44 V vs. NHE) which should be high enough to oxidize 3-MeQ, and is also much higher that the reduction potential of N_3_^●^ (E^0^ = +1.33 V vs. NHE). Provided that the reduction potential of N_3_^●^ (E^0^ = +1.33 V vs. NHE) is high enough to oxidize 3-MeQ, one should expect similar oxidation products formed from the reaction of these two radicals with 3-MeQ.

Under conditions ensuring participation in the oxidation reaction of only SO_4_^●−^ (vide Materials and Methods), the transient absorption spectrum recorded at 10 μs after the electron pulse in aqueous solutions at pH = 7 and containing 0.1 mM 3-MeQ is characterized by two absorption maxima at λ_max_ = 375 and 480 nm and featureless absorption band which shows no distinct λ_max_ > 270 nm (Figure 2). This absorption spectrum was tentatively assigned by us to the product of one-electron oxidation of 3-MeQ by SO_4_^●−^. On the other hand, under conditions ensuring participation in the potential oxidation reaction of only N_3_^●^ (vide Materials and Methods), the transient absorption spectrum recorded at 3 μs after the electron pulse in aqueous solutions at pH = 7 and containing 0.1 mM 3-MeQ is characterized by a narrow and distinct absorption band with λ_max_ = 355 nm (Figure 2).

Due to depletion of the 3-MeQ ground state which absorbs at the spectral region < 380 nm (vide Appendix A) the observed absorption maxima in this spectral region were shifted towards shorter wavelengths in both systems studied (vide Appendix A). Despite this inconvenience, there is no doubt that reactions of 3MeQ with SO_4_^●−^ and N_3_^●^ lead to different primary transient products.

Interestingly, no reaction of (SCN)_2_^●−^ with 3-MeQ was observed in N_2_O-saturated aqueous solutions at pH 7 containing 1 mM 3-MeQ and 10 mM KSCN.

#### 2.1.2. Absorption Spectra Recorded after Reaction of N_3_^●^ and SO_4_^●−^ with 7-R-3-MeQ Derivatives

In order to check whether reactions of N_3_^●^ and SO_4_^●−^ lead again to different transient products, similar experiments were performed with 3-MeQ derivatives containing electron-withdrawing substituents (-CN, -CF_3_, and -F), and electron-donating substituent (-OCH_3_) in position 7 in the benzene moiety.

The transient absorption spectra resulting from the reaction of SO_4_^●−^ and recorded at 6–7 μs after the electron pulse in aqueous solutions at pH = 4 and containing 0.1 mM 7-R-3-MeQ derivatives with electron-withdrawing substituents are characterized by the similar features in comparison to the absorption spectrum recorded in the case of 3-MeQ. They consisted of two absorption bands with maxima located at λ = 390 and 480 nm and featureless absorption band which shows no distinct λ_max_ > 300 nm (Figure 3a). Again, the transient absorption spectra resulting from the reaction of N_3_^●^ and recorded at 6–8 μs after the electron pulse in aqueous solutions at pH = 7 and containing 0.1 mM 7-R-3-MeQ derivatives were characterized by similar features in comparison to the absorption spectrum recorded in the case of 3-MeQ. They were characterized by a narrow and distinct absorption band with λ_max_ ≈ 350–360 nm (Figure 3b).

This is solid evidence that transients produced in both systems possess the same nature, irrespectively of the electron-withdrawing substituent in the position 7, however, they are different depending on the nature of the reacting radical.

Interestingly, the transient absorption spectra resulting from the reaction of SO_4_^●−^ and N_3_^●^ with 7-OCH_3_-3-MeQ and recorded at 5 μs and 8 μs after the electron pulse, in aqueous solutions at pH 4.4 and 7, respectively and containing 0.1 mM 7-OCH_3_-3-MeQ are characterized by the similar spectral features (Figure 4a) unlike to the spectra recorded for 3-MeQ (vide Figure 2).

Similar spectral features were observed in the transient absorption spectrum resulting from the reaction of N_3_^●^ with 7-NH_2_-3-MeQ recorded at 10 μs after the electron pulse, in aqueous solutions at pH = 7 and containing 0.1 mM 7-NH_2_-3-MeQ (vide Appendix A).

### 2.2. Reactions of Tl^2+^ and CO_3_^●−^ with 7-OCH_3_-3MeQ–Absorption Spectra

#### 2.2.1. Absorption Spectra Recorded in Slightly Acidic and Neutral Solutions

In order to check whether the other one-electron oxidant (Tl^2+^) is able to oxidize 7-OCH_3_-3-MeQ, the transient absorption spectra were recorded in N_2_O-saturated aqueous solutions containing 0.1 mM 7-OCH_3_-3-MeQ and 5 mM TlCl. The transient absorption spectrum resulted from the reaction of Tl^2+^ with 7-OCH_3_-3-MeQ and recorded at 40 μs after the electron pulse, in aqueous solutions at pH 3.0 is characterized by the similar spectral features (Figure 4a) in comparison to the spectra resulted from the reaction of SO_4_^●−^ and N_3_^●^. Therefore, these absorption spectra were assigned to the product of one-electron oxidation of 7-OCH_3_-3-MeQ which most likely is the 7-OCH_3_-3-MeQ^●+^ or its deprotonated neutral form 7-OCH_3_-3-MeQ^●^ (vide Discussion).

#### 2.2.2. Absorption Spectra Recorded in Alkaline Solutions

In the investigated pH range (3 to 11.3) 7-R-3-MeQ quinoxalin-2-ones exist in two forms (Appendix A) that are involved in acid-base equilibria with the respective *pK_a_* values located between pH 8.3 and 10.0 (Appendix A). In another words, in slightly acidic or neutral solutions the protonated form dominates whereas in alkaline solutions the anionic form.

Experiments, analogous to those in slightly acidic or neutral solutions, were performed in alkaline solutions in order to check whether the transient absorption spectra from reactions of anionic form of 7-OCH_3_-3-MeQ with N_3_^●^ as compared with another one-electron oxidant (CO_3_^●−^) were characterized by the similar spectral features. The transient absorption spectra resulting from the reaction of N_3_^●^ and CO_3_^●−^ and recorded at 9 and 50 μs, respectively after the electron pulse in aqueous solutions at pH 11.3 and containing 0.1 mM 7-OCH_3_-3-MeQ are characterized by the similar features. They consisted of two absorption bands with maxima located at λ = 420 and 540 nm and the third absorption band with λ_max_ = 320 nm (Figure 4b).

### 2.3. Reactions of N_3_^●^ with 3-Methyl-1H-Quinoxalin-2-one (3-MeQ) and Its 7-Substituted Derivatives (7-R-3MeQ–Kinetics

#### 2.3.1. Kinetics Recorded in Solutions Containing 3-MeQ and 7-R-3-MeQ Derivatives with Electron-Withdrawing Substituents

Experiments performed in N_2_O-saturated aqueous solutions containing 0.1 mM 3-MeQ and 7-R-MeQ (R = −CN, −CF_3_, −F) and 0.1 M NaN_3_ at pH 7 revealed that the spectral characteristics of the absorption spectra do not change with the substituent (vide Figure 2 and Figure 3b). They are characterized by a single absorption band with λ_max_ ≈ 350–360 nm. In order to determine directly the bimolecular rate constants of N_3_^●^ with 3-MeQ and 7-R-MeQ, a kinetic analysis at various concentration of 3-MeQ and 7-R-MeQ (0.05 mM–0.5 mM) was performed. The growth kinetics were recorded at λ = 370 nm. The rate of formation, followed at that wavelength fits to a single exponential (Figure 5a,b).

The pseudo-first-order rate constants of the formation of the 370-nm absorption bands were plotted as a function of 3-MeQ and 7-CN-3-MeQ concentrations (Figure 6). It is clearly seen that the pseudo-first-order rate constants measured at λ = 370 nm show a linear dependence on the concentration of 3-MeQ and 7-CN-3-MeQ in the full range of concentration studied. The slopes represent the second-order rate constants for the formation of transient resulting from the reaction of N_3_^•^ with 3-MeQ and 7-CN-3-MeQ. Interestingly, the linear plots have non-zero intercepts that indicate the involvement of equilibria (vide Discussion) and that represent first-order rate constants for the backward reactions in the equilibria.

The obtained values of *k*_forward_ and *k*_backward_ reactions together with the respective equilibrium constants (*K*) for 3-MeQ and 7-R-MeQ (R = −CN, −CF_3_, −F) derivatives are collected in Table 1.

Moreover, the maximum value of the 370-absorbance was dependent on the 3-MeQ and 7-CN-3-MeQ concentration. When this was increased for 3-MeQ from 0.05 mM to 0.5 mM, *G* × ε increased from 4.3 × 10^−4^ dm^3^ J^−1^ cm^−1^ to 12.6 × 10^−4^ dm^3^ J^−1^ cm^−1^ (vide Figure 5a). Analogously for 7-CN-3-MeQ from 0.1 mM to 0.5mM, *G* × ε increased from 13.6 × 10^−4^ dm^3^ J^−1^ cm^−1^ to 46.8 × 10^−4^ dm^3^ J^−1^ cm^−1^ (vide Figure 5b). These increases cannot be accounted for by the increase in *G*(N_3_^●^) scavenged due to the higher 3-MeQ and 7-CN-3-MeQ concentrations. It rather points again to the existence of an equilibrium situation presented in Scheme 2 (vide Discussion) where N_3_^●^ adducts were formed at the C2 carbon atom. These adducts were responsible for the absorption at λ = 370 nm.

#### 2.3.2. Kinetics Recorded in Neutral Solutions Containing 7-OCH_3_-3-MeQ

Experiments performed in N_2_O-saturated aqueous solutions containing 0.1 mM 7-OCH_3_-3-MeQ and 0.1 M NaN_3_ at pH 7 revealed that the absorption spectra were characterized by two absorption bands with maxima located at λ = 430 nm and 530 nm (vide Figure 4a). In order to directly determine the bimolecular rate constant of N_3_^●^ with 7-OCH_3_-3-MeQ, a kinetic analysis at various concentrations of 7-OCH_3_-3-MeQ (0.05 mM–0.5 mM) was performed. The growth kinetics followed at these wavelengths fit to a single exponential (Figure 7a and Appendix A).

The pseudo-first-order rate constants of the formation of the 430-nm and 530-nm absorption bands were plotted as a function of 7-OCH_3_-3-MeQ concentration (Figure 7b). Surprisingly, it is clearly seen that the pseudo-first-order rate constants measured at both wavelengths do not show a linear dependence on the concentration of 7-OCH_3_-3-MeQ in the range 0.05 mM–0.5 mM. Moreover, since they do not show any specific trend and considering the errors in the measurements of absorbencies and rate constants based on relatively weak signals, one can reasonably assume that they do not depend on the concentration of 7-OCH_3_-3-MeQ at all. The averaged values of the first-order rate constants are nearly equal: 3.8 × 10^5^ s^−1^ and 4.2 × 10^5^ s^−1^ for 430 and 530 nm, respectively (Figure 7b).

These results support the tentative hypothesis that the averaged first-order rate constant (≅4.0 × 10^5^ s^−1^) can be attributed to the secondary intramolecular process leading to the transient which most likely is the 7-OCH_3_-3-MeQ^●+^ or its deprotonated neutral form 7-OCH_3_-3-MeQ^●^ (vide Scheme 3 in Discussion).

#### 2.3.3. Kinetics Recorded in Alkaline Solutions Containing 7-OCH_3_-3-MeQ

Experiments performed in N_2_O-saturated aqueous solutions containing 0.1 mM 7-OCH_3_-3-MeQ and 0.1 M NaN_3_ at pH 11.3 revealed that the absorption spectra are characterized by two absorption bands with maxima located at λ = 430 nm and 530 nm (vide Figure 4b). Similarly, as for pH 7, in order to directly determine the bimolecular rate constant of N_3_^●^ with 7-OCH_3_-3-MeQ (however in the anionic form), a kinetic analysis at various concentrations of 7-OCH_3_-3-MeQ (0.05 mM–0.5 mM) was performed. The rate of formation, followed at these wavelengths fits to a single exponential (Figure 8a and Appendix A).

The pseudo-first-order rate constants of the formation of the 430-nm and 530-nm absorption bands were plotted as a function of 7-OCH_3_-3-MeQ concentration (Figure 8b). In this case, contrary to pH 7, it is clearly seen that the pseudo-first-order rate constants measured at both wavelengths show a linear dependence on the concentration of 7-OCH_3_-3-MeQ in the full range of concentration studied. The slope representing second-order rate constant for the forward reaction is equal to *k*_f_ = 5.5 × 10^9^ M^−1^ s^−1^ calculated at both wavelengths. Both linear plots have the intercepts indicating involvement of an equilibrium (vide Scheme 3 in Discussion) and representing the first-order rate constant for the backward reactions *k*_b_ = 9.7 × 10^4^ s^−1^ and 9.0 × 10^4^ s^−1^ calculated at 430-nm and 530-nm, respectively.

### 2.4. Theoretical Calculations of Radical Cations and Their Deprotonated Forms Derived from 3-MeQ and 7-R-3-MeQ Derivatives

Taking into account that SO_4_^●−^ is the strongest one-electron oxidant used in our studies one can reasonably expect that the main products of the 3-MeQ and 7-R-3-MeQ derivatives oxidation are the respective 3-MeQ^●+^/7-R-3-MeQ^●+^ and/or their deprotonated neutral forms 3-MeQ^●^/7-R-3-MeQ^●^. The calculated UV-vis spectra at the DFT level for the 3-MeQ^●+^/7-CN-3-MeQ^●+^ and their deprotonated neutral forms 3-MeQ^●^/7-CN-3-MeQ^●^ are presented in Figure 9.

Unfortunately, similarity between calculated spectra of 3-MeQ^●+^ and 3-MeQ^●^ does not allow for an unambiguous statement to which of these species can be assigned the experimentally observed spectrum (Figure 9a). The situation is somewhat clearer in the case of calculated spectra for the radical cations (7-CN-3-MeQ^●+^ and 7-OCH_3_-3-MeQ^●+^) and radicals (7-CN-3-MeQ^●^ and 7-OCH_3_-3-MeQ^●^) derived from 7-CN-3MeQ and 7-OCH_3_-3-MeQ^,^ respectively. The comparison of the calculated spectra with the experimental spectra leads to assigning the experimental spectra to 7-CN-3-MeQ^●^ (Figure 9b) and the 7-OCH_3_-3-MeQ^●^ (vide Appendix A).

### 2.5. Theoretical Calculations of N_3_^●^ Adducts to 3-MeQ and 7-R-3-MeQ Derivatives

In order to confirm the structure of the potential N_3_^●^ adducts to 3-MeQ and 7-R-3-MeQ derivatives, we considered two potential sites of N_3_^●^ addition, namely addition at the C-2 and C-3 carbon atoms in pyrazine moiety (vide Appendix A). The calculated UV-vis spectra at the DFT level for the respective adducts in 3-MeQ are presented in Figure 10.

A reasonably good agreement is observed between the experimental spectrum and the calculated spectrum of the adduct at the C2 carbon atom, particularly regarding a distinct absorption band with λ_max_ = 355 nm. An additional band with λ_max_ = 540 nm is almost invisible in the experimental spectrum, which is probably related to its very low intensity being at the limit of detection of the experimental system. This observation means that addition of N_3_^●^ to the double bond between the N1 nitrogen atom and the C2 carbon atom in pyrazine ring can only occur in the enolic tautomeric form of 3-MeQ (vide right side of the equilibrium in Figure 11). It has to be noted that the spectral features of the calculated spectrum of the adduct at the C3 carbon atom are not present in the experimental spectrum. For unknown reasons, addition of N_3_^●^ to the double bond between the C3 carbon atom and the N4 nitrogen atom in pyrazine ring does not seem to take place.

An additional proof that addition of N_3_^●^ can only occur at the C2 carbon atom in the enolic tautomeric form of 3-MeQ was obtained by performing experiments with 1,3-dimethyl-quinoxalin-2-one (1,3-diMeQ) where tautomeric equilibrium is not possible (Figure 11). In this chemical system no absorption band was observed in the 300–700 nm region which is a strong premise that addition at the C2 carbon atom is crucial for N_3_^●^. Interestingly, even in this case, addition of N_3_^●^ at the C3 carbon atom was not observed.

## 3. Discussion

### 3.1. Reaction Pathways Involving One-Electron Oxidants and 3-MeQ/7-R-3-MeQ Derivatives

Transient absorption spectra observed on reaction of SO_4_^●−^ with 3-MeQ (R = –H) and 7-R-3-MeQ derivatives containing electron-withdrawing substituents (−CN, −F, −CF_3_) are characterized by the similar features, i.e., two absorption bands with λ_max_ = 390 and 480 nm (vide Figure 2 and Figure 3a). It means that the character of electron-withdrawing substituents does not have an influence on the location of maxima of absorption bands. On the contrary, the absorption spectrum observed on reaction of SO_4_^●−^ with 7-OCH_3_-3-MeQ, though is characterized by two similar absorption bands, however, with maxima shifted towards longer wavelengths, λ_max_ = 420 and 540 nm (vide Figure 4a). The similar spectrum was observed on reaction of two other one-electron oxidants, i.e., Tl^2+^ and CO_3_^●−^ (vide Figure 4a,b). Since SO_4_^●−^, Tl^2+^, and CO_3_^●−^ are considered as one-electron oxidants with high or reasonably high reduction potentials (vide Appendix A), these spectra could in principle be assigned to either 7-R-3-MeQ^●+^ and/or their deprotonated forms (7-R-3-MeQ^●^). These radicals can be formed by a direct electron transfer (outer-sphere electron transfer) (Scheme 1).

Generally, the p*K*_a_ values of radicals are lower than the p*K*_a_ values of the compounds they are derived from. The ∆p*K*_a_ reflects the increase in acidity on one-electron oxidation. Such increase in acidity (with the 12 orders of magnitude) was observed for phenol (PhOH) (p*Ka* = 10.0) and phenol radical cation (PhOH^●+^) (pK_a_ = −2) [78] and, however smaller, for guanosine (G) (p*K*_a_ = 9.4) and guanosine radical cation (G^●+^) (p*K*_a_ = 3.9) [79]. The pK_a_ values of the acid-base equilibria of indolyl radical cations (InH^●+^) and N-centered indolyl radicals (In^●^) derived from indoles and tryptophan are another relevant examples. These values are located between 4.3 and 6.1 [10,11,80], and are much lower than the p*K*_a_ value of indole equal to 16.97 [81]. Therefore, there is no doubt that the transient formed by a direct electron transfer between CO_3_^●−^ with 7-OCH_3_-3-MeQ (present in deprotonated form at pH 11.3) is N1-centered radical, 7-OCH_3_-3-MeQ^●^ (Scheme 1). The question that arises at this point concerns the character of the transients which are formed at pH below p*K*_a_ of 7-R-3-MeQ compounds, where they exist in protonated forms (vide Appendix A). To answer this question, we compared the spectra observed on reaction of SO_4_^●−^ with 7-OCH_3_-3-MeQ at pH 4 and pH 7 (vide Appendix A) with the spectrum observed on reaction of CO_3_^●−^ with 7-OCH_3_-3-MeQ at pH 11.3 (vide Figure 4b). The same position of the absorption maxima in all three spectra is the first indication that 7-OCH_3_-3-MeQ^●^ is also present at pH 4 and 7. This radical is formed by deprotonation of 7-OCH_3_-3-MeQ^●+^ (Scheme 1). Generally, the radical cations absorb at longer wavelengths in comparison to their deprotonated forms. Again, spectra of indolyl radical cations and N-centered indolyl radicals derived from indoles and tryptophan are relevant examples [11,82]. This trend was also confirmed by the ωB97XD/aug-cc-pVTZ calculated UV-Vis spectra of the 7-OCH_3_-3-MeQ^•+^ and 7-OCH_3_-3-MeQ^•^ species (vide Appendix A). With these results and premises in hands, the molar absorption coefficients of transients at the respective maxima of spectra (λ = 420 and 540 nm) were calculated to be ε_420_ = 4000 M^−1^ cm^−1^ and ε_420_ = 4000 M^−1^ cm^−1^ for pH 4 and 7 taking the *G*-value of (SO_4_^●−^) = 0.29 μM J^−1^ and ε_420_ = 4100 M^−1^ cm^−1^ and ε_420_ = 3900 M^−1^ cm^−1^ for pH 11.3 taking the *G*-value of (CO_3_^●−^) = 0.61 μM J^−1^. The *G*-values of SO_4_^●−^ and CO_3_^●−^ for the respective concentrations of S_2_O_8_^2−^ and CO_3_^2−^ were calculated from the Schuler formula which allows correction of the *G*-values of their precursors (e^−^_aq_, HO^●^) resulting from competition between their scavenging reactions and track recombination processes [83,84]. These calculations confirm ultimately that 7-OCH_3_-3-MeQ^●^ is present at pH 4 and 7 which places the p*K*_a_ value of the acid-base equilibrium of 7-OCH_3_-3-MeQ^●+^ and 7-OCH_3_-3-MeQ^●^ below 4. This is consistent with the results obtained for guanosine (G), taking into account similar p*K*_a_ values of G and 7-OCH_3_-3-MeQ in the native state.

### 3.2. Reaction Pathway Involving N_3_^●^ and 3-MeQ/7-R-3-MeQ Derivatives with Electron Withdrawing Substituents (R)

Experimental observations that reactions of N_3_^●^ with 3-MeQ and 7-R-3-MeQ (R = −CN, −CF_3_ and −F) lead to significantly different absorption spectra (vide Figure 2 and Figure 3b) compared to the absorption spectra obtained in reactions with SO_4_^●−^ as one-electron oxidant (vide Figure 2 and Figure 3a) clearly indicate that electron transfer does not occur. This is due to the fact that N_3_^●^ is too weak oxidant be able to oxidize 3-MeQ and its derivatives with electron withdrawing substituents. In another words, the reduction potentials of these compounds must be higher than +1.33 V vs. NHE.

Thus, what is the identity of the transient species characterized by a distinct single absorption band with λ_max_ = 355–365 nm in solutions containing 3-MeQ and 7-R-3-MeQ (R = −CN, −CF_3_ and −F) at pH 7 (vide Figure 2 and Figure 3b)? The molecular structure of the respective tautomers, the lack of a similar absorption band in the presence of 1,3-dimetylquinoxalin-2-one and the comparison of calculated absorption bands of the potential N_3_^●^ adducts with the experimental spectra clearly show that the primary N_3_^●^ attack occurs only on a double C=N bond at the C2 carbon atom in the enolic tautomer. This observation suggests regioselectivity in the N_3_^●^ addition and in a consequence formation of N-centered radical on the N1 nitrogen atom (Scheme 2). These reactions occur with high-rate constants which ranged from 6.1 × 10^9^ M^−1^ s^−1^ to 9.8 × 10^9^ M^−1^ s^−1^ suggesting that they are nearly diffusion-controlled. Moreover, the interesting finding is reversibility of these reactions with an involvement of an equilibrium with *K*_eq_ ranged from 2.9 × 10^3^ M^−1^ to 1.2 × 10^4^ M^−1^ (Table 1).

### 3.3. Reaction Pathways Involving N_3_^●^ and 7-OCH_3_-3-MeQ

In the case of 7-OCH_3_-3-MeQ, the absorption spectrum of the transient formed by reaction with N_3_^●^ at pH 7 is similar to absorption spectra formed by reaction with SO_4_^●−^ and Tl^2+^ (vide Figure 4a). This spectrum is characterized by similar features in comparison to absorption spectra observed for the other 7-R-3-MeQ derivatives, except for a small shift of two absorption maxima towards longer wavelengths (compare Figure 3a and Figure 4a). On the basis of the previous assignments for 7-R-3-MeQ derivatives and the ωB97XD/aug-cc-pVTZ calculated UV-Vis spectra (vide Appendix A), the spectra presented on Figure 4a were assigned to 7-OCH_3_-3-MeQ^●^. Furthermore, absorption spectrum with similar features and consisted of two absorption bands with maxima located at λ = 460 and 560 nm was assigned to 7-NH_2_-3-MeQ^●^ formed by reaction of N_3_^●^ with 7-NH_2_-3-MeQ (vide Appendix A). These observations clearly indicate that N_3_^●^ are able to oxidize 7-OCH_3_-3-MeQ and 7-NH_2_-3-MeQ and impose values of their reduction potentials lower than +1.33 V vs. NHE. This is in line with the expected decrease of reduction potentials of 7-R-3-MeQ derivatives with electron-donating substituents.

Lack of the expected linear dependence of the pseudo-first-order rate constants of the formation of 7-OCH_3_-3-MeQ^●^ on concentration of 7-OCH_3_-3-MeQ at pH 7 requires explanation. The following mechanism is proposed to explain the experimental observations (Scheme 3).

The initial attack of N_3_^●^, first step, is by addition to the double bond C=N at the C2 carbon atom in the enolic tautomer producing N-centered radical on the N1 nitrogen atom. This radical undergo elimination of N_3_^−^ anion (“inner-sphere” electron transfer) leading to the 7-OCH_3_-3-MeQ^●+^ (second step) which further undergoes deprotonation leading to the 7-OCH_3_-3-MeQ^●^ (third step). Since the rate of formation of 7-OCH_3_-3-MeQ^●^ does not depend on the concentration of 7-OCH_3_-3-MeQ, the first step does not control its formation. The averaged value of the first-rate constant (*k* ≅ 4.0 × 10^5^ s^−1^) measured in the full range of concentration studied represents rather the rate of the second step which does not depend on 7-OCH_3_-3-MeQ concentration.

In the case of 7-OCH_3_-3-MeQ, the absorption spectrum of the transient formed by reaction with N_3_^●^ at pH 11.3 is similar to the absorption spectrum formed by reaction with CO_3_^●−^ (vide Figure 4b). Therefore, this spectrum was assigned to 7-OCH_3_-3-MeQ^●^. These observations clearly indicate that N_3_^●^ are able to oxidize 7-OCH_3_-3-MeQ in deprotonated state, i.e., 7-OCH_3_-3-MeQ^−^.

Moreover, it is interesting to note that at pH 11.3 the pseudo-first-order rate constants of the formation of 7-OCH_3_-3-MeQ^●^ depend linearly on concentration of 7-OCH_3_-3-MeQ (vide Figure 8b). The second order rate constant *k*_f_ = 5.5 × 10^9^ M^−1^ s^−1^ and the first order rate constants *k*_b_ = 9.7 × 10^4^ s^−1^ and 9.0 × 10^4^ s^−1^ represent the respective rate constants of forward and backward reactions in the equilibrium displayed at the bottom of Scheme 3. These values enable us to calculate the equilibrium constant of the electron transfer equilibria equal to *K*_eq_ = 5.67 × 10^4^ and 6.11 × 10^4^, respectively. Based on the simplified equation ∆E(mV) ≈ 59.1 × log *K*_eq_ [85], where ∆E = E^0^(N_3_^●^/N_3_^−^) − E^0^(7-OCH_3_-3-MeQ^●^/7-OCH_3_-3-MeQ^−^) one can easily estimate the reduction potential of 7-OCH_3_-3-MeQ^●^/7-OCH_3_-3-MeQ^−^ redox couple equal to +1.05 V vs. NHE. To our best knowledge, this is the first measurement of the reduction potential for one of quinoxalin-2-one derivatives, though at high pH. Thus, at pH larger than the p*K*_a_ of 7-OCH_3_-3-MeQ (Appendix A and Appendix A) N_3_^●^ react with 7-OCH_3_-3-MeQ^−^ by direct electron transfer (“outer-sphere” electron transfer).

## 4. Materials and Methods

### 4.1. Chemicals

Sodium azide (NaN_3_) (≥99.5% purity), sodium persulfate (Na_2_S_2_O_8_) (≥99% purity), thallium chloride (TlCl) (≥98% purity), sodium chloride (NaCl) (≥99.5% purity), potassium thiocyanate (KSCN) (≥99% purity), sodium carbonate (Na_2_CO_3_) (≥99% purity), perchloric acid (HClO_4_) (70%, 99.999% purity), sodium hydroxide (NaOH) (≥98% purity), and *tert*-butanol ((CH_3_)_3_COH) (≥99.5% purity) were purchased from Sigma-Aldrich (St. Louis, MO, USA) and used without further purification. Nitrous oxide (N_2_O) > 98% and argon (Ar) BIP PlusX50S were from Messer (Warsaw, Poland).

### 4.2. Synthesis of 7-Substituted 3-Methyl-2(1H)-Quinoxalin-2-Ones

7-substituted 3-methyl-2(1H)-quinoxalin-2-ones (see Figure 1) were prepared by the classical reaction of the corresponding *o*-phenyldiamines (1 mmol) by adding dropwise methyl pyruvate (1.2 mmol) and trimethylamine (3 mmol) in ethanol. A detailed description of the synthesis, purification, and spectral characterization were given elsewhere [55].

### 4.3. Preparation of Solutions

All solutions were made with water triply distilled provided by a Millipore Direct-Q 3-UV system. The pH was adjusted by the addition of NaOH or HClO_4_. Prior to irradiation, the samples were purged gently with N_2_O for 30 min. per 200 mL volume. The typical concentration of 7-substituted 3-methyl-2(1H)-quinoxalin-2-ones in solutions was 0.1 mM, unless otherwise specified. Experiments were performed with a continuous flow of sample solutions using a standard quartz cell with optical length 1 cm at room temperature (~23 °C).

### 4.4. Pulse Radiolysis Instrumentation

The pulse radiolysis experiments were performed with the LAE-10 linear accelerator at the Institute of Nuclear Chemistry and Technology in Warsaw, Poland with typical electron pulse length of 8 ns and 10 MeV of energy. A detailed description of the experimental setup has been given elsewhere along with basic details of the equipment and its data collection system [86,87]. The 150W xenon arc lamp E7536 (Hamamatsu Photonics K.K) and 1 kW UV-enhanced xenon arc lamp (Oriel Instruments) were alternately applied as monitoring light sources. The respective wavelengths were selected by MSH 301 monochromator (Lot Oriel Gruppe) with resolution 2.4 nm. The intensity of analyzing light was measured by means of PMT R955 (Hamamatsu). A signal from detector was digitized using a Le Croy WaveSurfer 104MXs-B (1 GHz, 10 GS/s) oscilloscope and then send to PC for further processing. Water filter was used to eliminate near IR wavelengths.

### 4.5. Pulse Radiolysis Experiments

#### 4.5.1. Dosimetry

The dosimetry was based on N_2_O-saturated solutions of 10^−2^ M KSCN which, following radiolysis, produces (SCN)_2_^●−^ radicals that have a molar absorption coefficient of 7580 M^−1^ cm^−1^ at λ = 472 nm and are produced with a yield of G = 0.635 µmol J^−1^ [88]. Absorbed doses per pulse were on the order of 20 Gy (1 Gy = 1 J kg^−1^).

#### 4.5.2. Selective Generation of the Primary Reactive Species

Pulse irradiation of Ar-saturated water leads to the formation of the primary reactive species shown in Equation (8):H_2_O ^^^^→ HO^●^, e^−^_aq_, H^●^(8)
with the following radiation chemical yields *G*(HO^●^) = 0.28 µmol J^−1^, *G*(e^−^_aq_) = 0.28 µmol J^−1^, and *G*(H^●^) = 0.06 µmol J^−1^ [89]. In N_2_O-saturated aqueous solutions hydrated electrons (e^−^_aq_) are converted into HO^●^ radicals (in the presence of H^+^) according to Equation (9) (*k*_9_ = 9.1 × 10^9^ M^−1^ s^−1^) [90] resulting in a corresponding increased yield of HO^●^ radicals.
e^−^_aq_ + N_2_O + H_2_O → N_2_ + HO^●^ + HO^−^(9)

In turn, in Ar-saturated aqueous solutions, HO^●^ radicals can be selectively removed by the addition of 2-methyl-2-propanol (*tert*-butanol) according to Equation (10) (k_10_ = 6.0 × 10^8^ M^−1^ s^−1^) [91], which is three orders of magnitude less reactive with hydrated electrons (*k* = 4 × 10^5^ M^−1^ s^−1^) [91].
HO^●^ + (CH_3_)_3_COH → H_2_O + ^●^CH_2_(CH_3_)_2_COH (10)

#### 4.5.3. Generation of the Selective Oxidizing Radicals and Metal Ions

*Azide radicals (N_3_^●^).* The azide radicals (N_3_^●^) were generated in N_2_O-saturated aqueous solutions containing 0.1 M NaN_3_ at pH 7, according to the Equation (11) (*k*_11_ = 1.2 × 10^10^ M^−1^ s^−1^) [91]:HO^●^ + N_3_^−^ → HO^−^ + N_3_^●^(11)

*Sulfate radical anions (SO_4_^●−^)*. The sulfate radical anions (SO_4_^●−^) were generated in Ar-saturated aqueous solutions containing 0.1 M K_2_S_2_O_8_ and 0.5 M of *tert*-butanol at pH 4, according to the Equation (12) (*k*_12_ = 1.2 × 10^10^ M^−1^ s^−1^) [91]:e^−^_aq_ + S_2_O_8_^2−^ → SO_4_^2−^ + SO_4_^●−^(12)

*Thallium cations (Tl^2+^)*. The thallium cations (Tl^2+^) were generated in N_2_O-saturated aqueous solutions containing 5 mM TlCl at pH 3, according to the Equation (13) (*k*_13_ = 9.9 × 10^9^ M^−1^ s^−1^) [91]:HO^●^ + Tl^+^ → HO^−^ + Tl^2+^(13)

*Carbonate radical anions (CO_3_^●−^)*. The carbonate radical anions (CO_3_^●−^) were generated in N_2_O-saturated aqueous solutions containing 0.1 M Na_2_CO_3_ at pH 11.3, according to the Equation (14) (*k*_14_ = 3.9 × 10^8^ M^−1^ s^−1^) [91].
HO^●^ + CO_3_^2−^ → HO^−^ + CO_3_^●−^(14)

*Dithiocyanate radical anions ((SCN)_2_^●−^)*. The dithiocyanate radical anions ((SCN)_2_^●−^) were generated in N_2_O-saturated aqueous solutions containing 0.01 M KSCN at pH 7, according to the Equation (15)–(17) (*k_f_*_15a_ = 1.4 × 10^10^ M^−1^ s^−1^) [92].
HO^●^ + SCN^−^ → HOSCN^●^(15)
HOSCN^●^^−^ ⇆ HO^−^ + SCN^●^(16)
SCN^●^ + SCN^−^ ⇆ (SCN)_2_^●^^−^(17)

### 4.6. Theoretical Procedures

All DFT [93] or TD-DFT [94] calculations, were they restricted or unrestricted, were performed using the dispersion correction ωB97XD functional [95] combined with the Dunning’s aug-cc-pVTZ correlation-consistent polarized basis sets augmented with diffuse functions [96,97] using the Gaussian 09 revision D.01 suite of programs [98]. The ωB97XD functional was recommended for calculation of many different properties [99,100,101,102], and the aug-cc-pVTZ basis set was among the most successful and widely used basis sets for post-Hartree-Fock and diverse DFT studies [101,103,104,105,106]. To ascertain that the optimized structures were true minima, the harmonic frequencies of all of them in the ground, radical and excited states were determined to be real. The charge, spin, and population analysis was conducted according to the NBO method [107] as implemented in the Gaussian 09 revision D.01 suite of programs [98]. Correlation analysis was done using the SigmaPlot 13 program [108] (version 13). The TD-DFT methods are known to predict spectra that are shifted from the experimental ones due to the basis set incompleteness and DFT functional inadequacy. Moreover, different transitions are often reproduced with a different shift. These factors always play a role especially when absorption spectra of transient radical species are calculated using a single reference method. Here, we manually shifted the calculated spectra by 20–130 nm to lower energies to best match the experimental ones.

## 5. Conclusions

In the current paper, we provided an experimental approach that allows to clearly define the nature of the obtained products formed in reactions of N_3_^●^ with quinoxalin-2-ones in aqueous solutions. This approach is based on comparison of transient absorption spectra observed on reaction of N_3_^●^ with those observed on reaction of one-electron oxidants (e.g., SO_4_^●−^, Tl^2+^ and CO_3_^●−^) with 3-MeQ and a series of 7-R-3-MeQ derivatives with electron withdrawing and electron-donating substituents. The mere fact that reactions of N_3_^●^ with quinoxalin-2-ones take place with high-rate constants (> 10^9^ M^−1^ s^−1^) does not ultimately prove that they are electron transfer reactions. For 3-MeQ and 7-R-3-MeQ derivatives with electron withdrawing substituents, the absorption spectra formed in reactions involving N_3_^●^ were different from the absorption spectra formed in the reactions involving other one-electron oxidants reacting only by electron transfer. Based on calculated absorption spectra employing density functional theory (DFT and TD-DFT), these spectra were assigned to N_3_^●^ adducts at the C2 carbon atom in pyrazine moiety. On the other hand, for 7-OCH_3_-3-MeQ the absorption spectra formed in reactions involving N_3_^●^ and other one-electron oxidants were similar and assigned to 7-OCH_3_-3-MeQ^●^. Interestingly, depending on pH, formation of 7-OCH_3_-3-MeQ^●^ can occur by either addition of N_3_^●^ to the pyrazine ring followed by elimination of SO_4_^●−^ (“inner-sphere” electron transfer) or by a direct electron transfer (“outer-sphere” electron transfer), and followed by deprotonation of 7-OCH_3_-3-MeQ^●+^. The approach presented here can be applied to other organic molecules containing double bonds for the proper assignment of absorption spectra either to N_3_^●^ one-electron oxidation products or its addition products. Such knowledge is important for evaluating the usefulness of N_3_^●^ as a secondary oxidant in biological studies in aqueous phase.

## Data Availability

All data is displayed in the manuscript.

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
