# Peer review of "Spectral Probe for Electron Transfer and Addition Reactions of Azide Radicals with Substituted Quinoxalin-2-Ones in Aqueous Solutions"

_ijms, 2021, doi:10.3390/ijms22020633_

Round 1
Reviewer 1 Report
"Spectral Probe for Electron Transfer and Addition Reactions of Azide Radicals with Substituted Quinoxalin-2-ones in Aqueous Solutions" is an excellent piece of work. I recommend that the manuscript should be accepted after a minor correction.
The phrase “N3● radical” must be replaced by “N3●” to avoid duplication. “N3● radical” means “N3 radical radical”.
Further review is not needed.
Author Response
We thank reviewer for high evaluation of the manuscript.
The phrase “N3● radical” must be replaced by “N3●” to avoid duplication. “N3● radical” means “N3 radical radical”.
Authors’ Reply:
We replaced the phrase “N3● radical/s” by “N3●” in the following places:
P.1, lines 20, 27, 29, 31, 32, 33, 35
P.2, lines 48, 52, 62, 66, 74, 76, 80
P.3, lines 88, 95, 97, 98, 100, 103, 106, 107, 113
P.4, lines 154, 164, 165
P.5, lines 172, 174, 175, 178, 183, 184, 193, 204
P.6, lines 218, 231
P.7, lines 241, 243, 252, 264, 268, 276
P.8, line 288
P.9, lines 311, 337
P11, lines 379, 392, 396
P.12, lines 400, 404, 405
P.13, lines 456, 458, 461, 470, 471
P.14, lines 480, 481, 483, 490, 491, 499, 501
P.15, lines 510, 512, 524
P.17, lines 607, 609, 611, 614, 617, 618, 620, 625
Reviewer 2 Report
In this article, the authors present an interesting study of the possible reaction mechanisms between azide radicals with substituted quinoxalin-2-ones in aqueous solutions. The formation of reaction intermediates is inferred by comparing the transient absorption spectra for the same reactions with other one-electron oxidizing radicals. Besides, theoretical calculations of some of those spectra are used to confirm the experimental conclusions. As my expertise is on the QM calculations, I will only comment on that theoretical part.
- On page 11, the authors state: ”The comparison of the calculated spectra with the experimental spectra leads to assigning the experimental spectra to the 7-CN-3-MeQ● radical (Figure 9b) and the 7-OCH3-3-MeQ● (vide Figure S7).” A more concrete explanation to justify that assignment is needed because it is not so evident from the differences in the spectra shown in both Figures.
- Is there a simple explanation for the more similarity of the calculated spectra for 3-MeQ●+ and 3-MeQ● ?
- In Figure 10, the agreement between the experimental and the calculated spectrum of the adduct at the C2 carbon atom is quite better than for the radicals referred to in 1). That is relevant because the formation of that adduct is a central point of this paper. However, this result must be validated by a plausible theoretical explanation following the one given in 1).
- The shifts of the calculated spectra seem rather arbitrary. They seem to have been included just to match better the experimental and the calculated results. Is that so?
- In the theoretical procedures, it is affirmed: “the harmonic frequencies of all of them in the ground, radical and excited states were determined to be positive.” Frequencies are real or imaginary but never positive. Eigenvalues of the force constants Hessian matrix are positive, negative, or zero.
Author Response
We thank reviewer for the positive evaluation of the manuscript.
- On page 11, the authors state:”The comparison of the calculated spectra with the experimental spectra leads to assigning the experimental spectra to the 7-CN-3-MeQ● radical (Figure 9b) and the 7-OCH3-3-MeQ● (vide Figure S7).” A more concrete explanation to justify that assignment is needed because it is not so evident from the differences in the spectra shown in both Figures.
Authors’Reply: The main reason to state that the experimental spectra are more similar to the calculated 7-CN-3-MeQ● than the 7-CN-3-MeQ+● (Figure 9b) is the distance between the two band maxima positioned above l = 350 nm. For the calculated radical the distance matches the experimental one while for the radical cation it is much larger because location of the absorption maximum of its band at longer wavelengths is much more red-shifted than that of the neutral radical. However, this is indeed not so evident for 7-OCH3-3-MeQ● (Figure S7). In Figure S7 we assumed the same shift of the calculated spectra for radical and radical cation which should not necessarily be correct. The shift corrections for radicals and radical cations may be different, and then spectra of the two species could fit the experimental spectra of the 7-OCH3-3-MeQ transient species equally well. However, for correctness of the assignment of the experimental spectra to the 7-CN-3-MeQ● we have also experimental arguments taken either from this work or from the literature which were presented in Discussion section in subchapter 3.1: P.12, line 425 to P.13, line 455.
- Is there a simple explanation for the more similarity of the calculated spectra for 3-MeQ●+ and 3-MeQ● ?
Authors’ Reply: The computational studies demonstrated that the 7-R-3-MeQ●+ and 7-R-3-MeQ● spectra are influenced by the substituent effect which affects particularly the absorption band at longer wavelengths more than the other bands. For the 7-H-3-MeQ the substituent effect is negligible (Figure 9a), while for the 7-CN-3-MeQ (Figure 9b) and 7-OCH3-3-MeQ (Figure S7), the absorption bands of radical cations are much more red-shifted then for neutral radicals. In the ground singlet state, the -CN and -OCH3 substituents are s electrons withdrawing but =OCH3 is p-electrons donating but CN is p-electrons withdrawing (vide e.g. Table 2, in J. Phys. Org. Chem. 2009, 22, 769-778 paper “σ- and π-electron contributions to the substituent effect: natural population analysis” by W. P. Ozimiński, J. Cz. Dobrowolski). However, unexpectedly, for the doublet radical cations, the -CN substituent changes its character: like the -OCH3 substituent, it is s-electrons withdrawing but p-electrons donating ! (J. Cz. Dobrowolski, G. Karpińska in “The σ- and π-electron contributions to the Substituent Effect in the Cation Radicals of Monosubstituted Benzenes”; paper which will be submitted in 2021). This is why, qualitatively, the spectra characteristics for radicals and radical cations of 7-CN-3-MeQ and 7-OCH3-3-MeQ look very similar.
- In Figure 10, the agreement between the experimental and the calculated spectrum of the adduct at the C2 carbon atom is quite better than for the radicals referred to in 1). That is relevant because the formation of that adduct is a central point of this paper. However, this result must be validated by a plausible theoretical explanation following the one given in 1).
Authors’ Reply: The experimental spectrum presented in Figure 10 agrees with that calculated for the N3• adduct at the C2 carbon atom whereas disagrees with that calculated for the N3• adduct at the C3 carbon atom. First, this is seen from the relative band intensities which are quite different for the experimental spectrum and those calculated for the C3 adduct, while they are quite similar for the C2 adduct. Second, the position of the lowest energy band in the C2 adduct and experimental system is congruent while for the C3 adduct is not.
- The shifts of the calculated spectra seem rather arbitrary. They seem to have been included just to match better the experimental and the calculated results. Is that so?
Authors’ Reply: That is right. The shifts were arbitrary just to emphasize the similarities in shape. Rigorous considerations determining all factors that influence band position of excited, short-lived, transient species, are hardly possible using approximate single-reference DFT method. We have commented on this issue in the Computation section.
- In the theoretical procedures, it is affirmed: “the harmonic frequencies of all of them in the ground, radical and excited states were determined to be positive.” Frequencies are real or imaginary but never positive. Eigenvalues of the force constants Hessian matrix are positive, negative, or zero.
Authors’ Reply: We thank reviewer for his suggestion. The word 'positive' was replaced by 'real'.